# A Novel Mast Cell Stabilizer JM25-1 Rehabilitates Impaired Gut Barrier by Targeting the Corticotropin-Releasing Hormone Receptors

**DOI:** 10.3390/ph16010047

**Published:** 2022-12-29

**Authors:** Yueshan Sun, Hong Li, Lei Liu, Xiaoqin Bai, Liping Wu, Jing Shan, Xiaobin Sun, Qiong Wang, Yuanbiao Guo

**Affiliations:** 1Medical Research Center, The Affiliated Hospital of Southwest Jiaotong University, The Third People’s Hospital of Chengdu, Chengdu 610031, China; 2Laboratory of Ethnopharmacology, Tissue-Orientated Property of Chinese Medicine Key Laboratory of Sichuan Province, West China School of Medicine, West China Hospital, Sichuan University, Chengdu 610065, China; 3Digestive Department, The Affiliated Hospital of Southwest Jiaotong University, The Third People’s Hospital of Chengdu, Chengdu 610031, China

**Keywords:** JM25-1, mast cell, intestinal barrier, CRHR, PI3K/AKT/mTOR

## Abstract

Mast cell (MC) plays a central role in intestinal permeability; however, few MC-targeting drugs are currently available for protection of the intestinal barrier in clinical practice. A nonfluorinated Lidocaine analog 2-diethylamino-N-2,5-dimethylphenyl acetamide (JM25-1) displays anti-allergic effect, but its impact on MC remains elusive. In this study, we explored whether JM25-1 has therapeutic potential on intestinal barrier defect through stabilizing MC. JM25-1 alleviated release of β-hexosaminidase and cytokine production of MC. The paracellular permeability was redressed by JM25-1 in intestinal epithelial cell monolayers co-cultured with activated MC. In vivo, JM25-1 diminished intestinal mucosal MC amount and cytokine production, especially downregulating the expression of CRHR1, accompanied by an increase of CRHR2. Protective effects appeared in JM25-1-treated stress rats with a recovery of weight and intestinal barrier integrity. Through network pharmacology analysis, JM25-1 showed a therapeutic possibility for irritable bowel syndrome (IBS) with predictive targeting on PI3K/AKT/mTOR signaling. As expected, JM25-1 reinforced p-PI3K, p-AKT, p-mTOR signaling in MC, while the mTOR inhibitor Rapamycin reversed the action of JM25-1 on the expression of CRHR1 and CRHR2. Moreover, JM25-1 successfully remedied intestinal defect and declined MC and CRHR1 expression in rat colon caused by colonic mucus of IBS patients. Our data implied that JM25-1 possessed therapeutic capacity against intestinal barrier defects by targeting the CRH receptors of MC through PI3K/AKT/mTOR signaling.

## 1. Introduction

The intestinal mucosa barrier plays a crucial role in preventing noxious contents of the intestinal lumen, in order to maintain systemic homeostasis [1]. The disruption of the epithelial barrier will increase intestinal permeability, leading to leaky gut syndrome [2]. Evidently, barrier defect is a vital pathogenesis of many diseases, mainly affecting the gut, such as inflammatory bowel disease (IBD) and irritable bowel syndrome (IBS), as well as systemic disease of other organ systems, e.g., type I diabetes and autism [3]. So far, many relevant therapeutic approaches for preventing intestinal permeability and barrier dysfunction have been explored such as nutrient assimilation, MC stabilizers, muco-protectors, epigenetic and exosome-mediated regulators of intestinal barrier function, and so on [4,5]. Yet there are no FDA-approved or even investigational drugs available targeting gut barrier permeability [3,6].

Accumulating evidence indicates that MC activated by CRH is a vital player in stress-induced enhanced intestinal permeability [7,8,9]. In detail, chronic stress triggers the activation of the hypothalamic-pituitary-adrenal axis (HPA) and increase CRH secretion; subsequently intestinal MC is activated by CRH and releases mediators, including tryptase, histamine, serotonin, chymase and inflammatory factors (IL-6, IL-1β, TNFα, etc.), which impair the intracellular tight junction (TJ) between the intestinal epithelial cells [7,10,11]. CRH receptor subtype 1 (CRHR1) and subtype 2 (CRHR2) on MC are positive and negative modulators, respectively, tuning the degranulation of stress-induced MC [12,13].Psychological stress is a common cause and induces a long-term depressive symptom for IBS, a disorder with an increase of MC and accompanied by intestinal dysbiosis in the gastrointestinal tract [14,15,16,17]. Certainly, targeting MC is one of the main approaches for dysregulated mucosal permeability [5].

JM25-1, a nonfluorinated lidocaine analog with limited anesthetic activity, has an inhibitory effect on bronchospasm and airway inflammation by decreasing lung eosinophil, neutrophil and lymphocyte proliferation [18]. Aerosolized JM25-1 also has an inhibitory effect on allergen-induced inflammation [18,19], but its effect on MC has not been well documented until now. Since activation of MC contributes to various forms of allergic diseases [20], in this study we explored the effects of JM25-1 on the activation of MC and its therapeutic potential against stress-induced intestinal permeability by in vitro and in vivo experiments.

## 2. Results

### 2.1. Effect of JM25-1 on HMC-1 and RBL-2H3 Cell Viability

To assess appropriate concentrations of JM25-1 in the following experiments, CCK-8 assay was admitted. As is shown, HMC-1 cells (Figure 1A) and RBL-2H3 (Figure 1D) were treated with different concentrations (3.9–250 μM) of JM25-1 for 24 h and there was no significant effect on proliferation of both the cell lines. Therefore, the concentration of JM25-1 below 250 μM was used to investigate the obstruction of MC degranulation.

### 2.2. JM25-1 Inhibits Degranulation in Mast Cell HMC-1 and RBL-2H3 Cells

When MC is activated by CRH or C48/80, secretory granules are filled with various preformed molecules such as lysosomal proteins, histamine, heparin and β-hexosaminidase, among others [21]. JM25-1 decreased the release of β-hexosaminidase from 7.8125 μM to 125 μM in C48/80-induced (Figure 1B) and in CRH-induced activation of HMC-1 cells (Figure 1C). Moreover, similar inhibition of JM25-1 was also found in C48/80- or CRH-induced degranulation on RBL-2H3 cells (Figure 1E,F).

### 2.3. JM25-1 Diminishes the Expression of Cytokines by Regulating the Expression of CRH Receptors in HMC-1 Cells

CRHR1 and CRHR2 expressed on MC act as a positive and a negative modulator of stress-induced MC degranulation, respectively [12]. We next explored the expression of CRHR and MC cytokines caused by JM25-1. As shown in Figure 2A,B, JM25-1 obviously suppressed both protein and mRNA levels of CRHR1 and TNFα while leading to elevated CRHR2 levels, compared to their levels in the untreated HMC-1 cells exposed to CRH. Meanwhile, treatment with JM25-1 decreased mRNA expression of IL-1β, IL-18 and IL-6 production of HMC-1 cells induced by CRH (Figure 2B). In addition, similar results were obtained in fluorescence images, in that JM25-1 significantly contributed to the expression of CRHR1 but suppressed CRHR2 (Figure 2C). Collectively, these data indicated that JM25-1 inhibited proinflammatory cytokine production in the activation of MC, including TNFα, IL-1β, IL-18 and IL-6, may be closely related to the regulation of CRH receptors. 

### 2.4. JM25-1 Restrains Epithelial Permeability in a MC-Dependent Way

Proinflammatory and regulatory mediators are released along with MC activation, and many of them have an effect on the intestinal barrier as well as modulating immune response [5,22]. With zona occludens 1 protein (ZO-1) as an indicator of the intercellular TJ between intestinal epithelial cells, we observed that TJ was markedly damaged on Caco2 cells by the culture medium of HMC-1 pre-treated with CRH, and JM25-1 restored the TJ (Figure 3A,D,E). The JM25-1 alone has no significant impact on ZO-1, which is shown in Appendix A. Additionally, transepithelial electrical resistance (TEER) of Caco2 cell monolayers is a common model used to investigate the permeability of the intestinal barrier [23]. The TEER value continuously dropped due to the co-culture with the medium of HMC-1 cells pre-treated with CRH. Nevertheless, JM25-1 treatment rebounded the TEER value closely to the control level (Figure 3B). This implied that JM25-1 prevented the permeability induced by the medium of activated-HMC-1 cells. Another means to evaluate barrier function is FD4 assay. Significant fluorescein penetrated the Caco2 layer after incubation with the medium of CRH-pretreated HMC-1 cells. Similarly, JM25-1 stopped the leaking of fluorescein (Figure 3C). Taken together, these results suggest that JM25-1 could maintain epithelial barrier integrity against the damage from the medium of activated-MC.

### 2.5. JM25-1 Protects the Intestinal Barrier from Psychological Stress

A growing amount of evidence has arisen from psychological stress-treated animal models, widely employed for studies of intestinal paracellular permeability [7,24,25]. To evaluate the effect of JM25-1 on rehabilitation of the epithelial barrier, psychological stress-induced rats were injected with JM25-1 intraperitoneally. There was a significant weight gain in the treated group compared to the non-treated group (Figure 4B). When the integrity of the intestinal barrier is damaged, diamine oxidase, D-lactate and bacterial endotoxin are released into the blood [26,27]. We monitored these markers in the blood from tail vein and found them increased in psychologically stressed rat on day 7 (Appendix A), and decreased when treated with JM25-1 on day 11 (Figure 4C). To further demonstrate that JM25-1 has a therapeutic effect on the intestinal epithelial barrier, TJ was detected with an immunofluorescence assay. As shown in the figures, both protein (Figure 4D) and mRNA (Figure 5A) level of ZO-1 were decreased in the small intestine of chronic stress-induced rats, whereas this damage was fixed by JM25-1 treatment. Overall, these results demonstrated that JM25-1 might renew the intestinal barrier impaired by chronic stress.

### 2.6. Effect of JM25-1 on the Expression of Intestinal CRH Receptors and Cytokines in Psychologically Stressed Rats

Psychological stress activates MC through regulation of CRHR1 and CRHR2, and leads to degranulation and release of cytokines or chemokines [7,17]; we then investigated whether these effects are administrated by JM25-1. With psychologically stressed rat models, JM25-1 has been shown to substantially inhibit CRHR1 (Figure 5B), TNFα (Figure 5D), IL-18 (Figure 5E), IL-1β (Figure 5F) and IL-6 (Figure 5G), but promoted CRHR2 (Figure 5C) production. Similar results were identified by immunofluorescence in MC marked with Tryptase. The amount of CRHR1-positive MC was significantly increased in the small intestine after stress stimulation, but was repressed after JM25-1 treatment (Figure 5H), and the TNFα production in intestinal tissue was also suppressed (Figure 5I). Taken together, JM25-1 affected the expression of intestinal CRH receptors and cytokines in psychologically stressed rats, which may be related to MC. 

### 2.7. Network Pharmacology Analysis of JM25-1 against IBS

To acquire the drug targets and the disease targets, we separately integrated the JM25-1-related targets obtained from SwissTargetPrediction with PharmMapper analyses and the IBS-associated targets obtained from GeneCards and DisGeNET. There are 110 drug targets and 3170 disease targets obtained after removing the duplicate targets. Through data comparison, 71 potential therapeutic targets for JM25-1 treatment of IBS were obtained. These targets are shown in Figure 6A.

To obtain hub targets, the protein–protein interaction of 71 therapeutic targets were analyzed through the STRING database. The criterion that the confidence score is greater than 0.4 was met and the desired target data was imported into Cytoscape 3.7.1 to generate an interactive network which includes 71 nodes and 205 edges. In a network, the more nodes, the more targets. The smaller degree value, the closer the color is to orange. Moreover, the larger the degree value, the closer the node color is to blue. Additionally, the edges among the nodes represented PPIs. The thickness of the line denoted the intensity of the protein–protein interaction (Figure 6B).

Then, a topology analysis of all nodes was performed using Cytoscape to investigate the hub targets. We set the degree value to greater than 13 for each node (target), then eight hub targets were obtained; namely, AKT1, SLC6A3, SRC, SLC6A4, SLC6A2, CHRNA4, DRD2 and GRIN2B were identified, as shown in Table 1. 

Then, molecular docking stimulation was used to estimate the binding abilities of JM25-1 to the hub target (AKT1). In this study, a docking score > 5.0 was deemed to indicate high binding activity [28]. As shown in Figure 6C, AKT1 exhibited strong affinities for JM25-1, with a docking score of 6.06.

To predict the signaling pathway, we conducted GO and KEGG analyses. The GO analysis results showed that a total of 331 GO terms were related to IBS virtual treatment with JM25-1. In the biological process category as shown in Figure 6D, 236 terms were mainly involved in response to drug and serotonin receptor signaling pathway, G-protein coupled receptor signaling pathway and protein phosphorylation. In the cellular component category, 38 terms were mainly involved in the plasma membrane, integral component of plasma membrane and cytoskeleton. In the molecular function category, 57 terms were mainly involved in serotonin binding, G-protein coupled serotonin receptor activity, protein kinase activity, drug binding and ATP binding. Moreover, the results of the KEGG analysis revealed that 62 pathways might be involved in JM25-1 regulatory action in IBS as shown in Figure 6E, mainly the mTOR signaling pathway, PI3K-AKT signaling pathway, ErbB signaling pathway, VEGF signaling pathway, Rap1 signaling pathway and Ras signaling pathway. The top 30 most-enriched pathways were arranged in ascending order of *p*-values.

### 2.8. JM25-1 Regulates CRHR Expression and Activation of MC through the PI3K/AKT/mTOR Signaling Pathway

As predicted, PI3K/AKT/mTOR pathway was possibly affected by JM25-1 on IBS. Next, we investigated the potential mechanism underlying JM25-1 on MC. Expectedly, JM25-1 facilitated phosphorylation of PI3K, AKT and mTOR in a concentration dependent manner (Figure 7A), suggesting that JM25-1 could activate the PI3K/AKT/mTOR signaling pathway. Additionally, a mTOR inhibitor Rapamycin was used on HMC-1 cells to further observe the necessity of this pathway for the action of JM25-1 on MC. As shown in the degranulation assay, Rapamycin successfully attenuated the interference of JM25-1 on MC activation (Figure 7B) and the secretion of cytokines tryptase, TNFα and IL-1β (Figure 7C). We also detected the protein and mRNA level of CRH receptors, and found that mTOR inhibitor prevented the effects of JM25-1 on the expression of CRHR1 and CRHR2 (Figure 7C,D). In conclusion, these data indicated that the effect of JM25-1 on degranulation and cytokine release of HMC-1 cells might depend on the PI3K/AKT/mTOR signal.

### 2.9. JM25-1 Regulated Intestinal Barrier Defect Induced by Colonic Mucus of IBS Patients

To investigate the value of clinical application, we further transplanted colonic mucus from IBS patients into rat colon through enema. Notably, JM25-1 treatment remarkably lowered the intestinal permeability induced by IBS mucus, as evidenced by a significant decrease in blood D-lactate (Figure 8A). In rat colonic segments, the TJ and microvilli ultrastructure had been destroyed by IBS mucus, but were refreshed with JM25-1 treatment (Figure 8B). Accordingly, the epithelial integrity indicating by ZO-1 protein was recuperated in the treatment group (Figure 8C). Besides, we found that JM25-1 significantly reduced the amount of MC and the expression of CRHR1 in MC in colon (Figure 8D).

## 3. Discussion

The loss of intestinal barrier is closely related to IBS, obesity, metabolic disorders, and so on [29], yet no therapy is available to recover the barrier, although a large quantity of literature has struggled to explore its mechanism and therapeutic strategies. Up to date, some bioactive pharmaceutical molecules such as epithelial cell death reducer, zonulin antagonist, muco-protectants, the blockers of CRH receptors, MC stabilizers and anti-inflammatory genes regulators were thought to be candidates [5].

Lidocaine as the local anesthetic most commonly used has been reported to have anti-tumor [30], anti-microbial [31], bronchiectasis [32], neuroprotective and anti-inflammatory effects [33]. However, if lidocaine were merely used for such functions, its fluorinated anilines, which conduct the initial anesthetic action, would lead to adverse effects such as hemolytic anemia, DNA damage and anesthesia itself [18,34,35]. Compared to lidocaine, a nonfluorinated analog JM25-1 with limited anesthetic activity was found to have more effective anti-inflammatory potential to inhibit bronchospasm and airway inflammation induced by allergic antigens [18,19]. Nevertheless, the effect of JM25-1 on the biological functions of MC is poorly understood. In this study, we found that JM25-1 inhibits CRH or C48/80-mediated activation of MC and alleviates the production of activated-MC and mediator in chronic stress-induced IBS rats. We considered that JM25-1 might be a new MC stabilizer or CRH receptor blocker to achieve its anti-inflammatory effects. 

Intestinal barrier defect is one of the most important routes of potential pathogenesis in IBS by regulating the activation of MC [36]. According to previous work, compared with healthy control, increased amount of MC is implicated, with documented alterations in the intestine among IBS patient [37,38]. Enhanced small intestinal permeability following psychological stress or exogenous CRH rely on MC since they are blocked by a MC stabilizer, sodium cromoglycate (DSCG) [7]. Given the inhibitory effect of JM25-1 on MC activation, we observed its treatment on intestinal permeability with in vitro and in vivo models. From the data on poly-glucosan permeability and TEER of epithelial cells, we demonstrated that JM25-1 protected the epithelial integrity against MC-induced damage. A treatment with JM25-1 successfully maintained the integrity of the intestinal barrier and relieved intestinal permeability in rat stress models. In addition, the medicated rats recovered their body weight and ameliorated the intestinal level of inflammatory mediators TNFα, IL-1β, IL-6 and IL-18. Together, JM25-1 regulates the release of mediators in MC to improve intestinal epithelial barrier function. 

MC activation and the consequent pathophysiologic responses to immunologic and psychological stressors are closely associated with down-regulation of CRHR1 and the simultaneously up-regulation of CRHR2 [12]. Early weaning stress on pigs showed that CRHR1 mediates intestinal permeability and hypersecretion, whereas CRHR2 has a protective property for the intestine barrier [39]. Clinical research also showed that diarrhea-predominant IBS (IBS-D) displayed jejunal up-regulation of CRHR2 and down-regulation of CRHR1 compared with healthy patients [13]. Non-selective antagonists of CRF receptor (such as Astressin) were repeatedly reported to reverse the stress-induced permeability in animal models, but the CRHR2-specific antagonist does not have this function [40,41,42]. CRHR2 and CRHR1 exist in a functionally antagonistic relationship [12]. The CRHR1-triggered inflammatory responses could be impeded by CRHR2 activation [40]. Furthermore, the stress-induced intestinal permeability could be hindered by activation of CRHR2 in MCs [12]. JM25-1 adopted the same pathway, that is, significantly promoting expression of CRHR2, while suppressing CRHR1 levels in MC. Accordingly, in the small intestine of chronic stress rats, JM25-1 significantly restrained the increase of CRHR1 in their mucosal MC. Importantly, JM25-1 successfully rebuilt the intestinal barrier damaged by IBS colonic mucus and reduced the MC amount and the expression of CRHR1, giving a preclinical implication for further development.

Further, we attempted to look for the pathway that is taken by JM25-1 to regulate CRH receptors. Network pharmacology has provided new perspectives in analyzing the relationship among drugs, targets and diseases. Employing this approach, we identified the relevant targets of JM25-1 against IBS and found PI3K-AKT and mTOR signaling pathways are likely to be involved in its drug actions, when PI3K activated AKT is recruited to the plasma membrane and phosphorylated and mTOR is positively regulated through the PI3K/AKT pathway [43]. According to previous research, neuropeptides that function to regulate stress responses could mediate the anti-inflammatory effect involved in PI3K/AKT and the glycogen synthase kinase-3β pathway [44]. Moreover, it was reported that MrgprX2 regulates MC degranulation through PI3K/AKT and PLCγ signaling in pseudo-allergic reactions [45]. Others have reported that the AKT phosphorylation plays a key role in vascular smooth muscle cells and regulates pro-inflammatory IL-6 secretion via CRHR2 [46]. Of interest, CRHR1/CRHR2 ligands activated AKT and CRHR1 signaling, and reduced apoptosis in human islets [47]. In this study, we demonstrated that JM25-1 contributed to the up-regulation of PI3K/AKT/mTOR pathway and CRHR2, but down-regulation of CRHR1 in MC activated by CRH. Utilizing Rapamycin, a mTOR inhibitor, we found that the signaling contributed to the down-regulation of β-hexosaminidase, Tryptase, TNFα, IL-1β release and CRHR1. Taken together, JM25-1 is a candidate for development of therapies against stress induced-intestinal barrier defect through MC via the PI3K/AKT/mTOR/CRHR pathway (Graphical Abstract). 

This study is the first to focus on the therapeutic potential of JM25-1 targeting intestinal barrier defects with IBS animal models or with the transplantation of IBS colonic mucus. Moreover, the mechanism underlying this effect was also documented, in that CRH receptors on mast cells were regulated by JM25-1 through the PI3K/AKT/mTOR signal pathway. In view of the above discussion, this research provides information on the novel pharmacological effects of JM25-1, which is valuable for development into new candidates for the treatment of barrier defect. However, there are some limitations in this study that deserve further improvement. On the one hand, future studies need to use MC knockout mice to verify that JM25-1 directly targets MCs. On the other hand, we intend to explore different administration methods, such as intragastric administration.

## 4. Materials and Methods

### 4.1. Cell Culture

HMC-1 cells (human mast cell line), RBL-2H3 (rat basophilic cell line) and Caco2 cells (epithelial-like cell line) were acquired from American Type Culture Collection (Rockville, MD, USA) and cultured in Iscove’s Modified Dulbecco’s Medium (IMDM, Hyclone, Shanghai, China) and RPMI-1640 (Hyclone, Shanghai, China), respectively, which were supplemented with 10% FBS and 1% antibiotics (100 U/mL penicillin and 100 mg/mL streptomycin), following incubation at 37 °C in a humidified atmosphere with 5% CO_2_. JM25-1 (Shandong Chengchuang Blue Sea Pharmaceutical Technology Co., Ltd., Shandong, China) was firstly dissolved in dimethyl sulfoxide (DMSO, MERK, Darmstadt, Germany), and configured as the stock solution, which was further diluted to 31.25 μM as a working solution with IMDM medium immediately before the experiments. Cells were treated with DMSO of the same volume in the drug group as Control group in vitro (≤0.1%).

### 4.2. Cell Cytotoxicity Assay

The cell viability was determined using a Cell Counting Kit-8 (CCK-8, Biosharp, Hefei, China). In brief, HMC-1 cells (1 × 10^4^ cells/well in 96-well plates) were treated with different concentrations of JM25-1 (3.9–250 μM) for 24 h. Ten microliters of CCK-8 was then incubated for another 2 h. The OD_450_ values were detected using Microplate Reader (PerkinElmer, Waltham, MA, USA), and the relative cell viability was calculated by the formula: [(OD _Treated_ − OD _Blank_)/(OD _Control_ − OD _Blank_)] × 100%. All experiments were performed at least three times in triplicate.

### 4.3. Integrity and Paracellular Permeability Assay

Caco2 cells (5 × 10^5^ cells/mL) and HMC-1 cells (5 × 10^5^ cells/mL) were seeded to the Transwell^®^ Flters (0.4 μm pore size, Corning, NY, USA) and basolateral compartment of the Transwells, respectively. After 21 to 23 days, Caco2 established a differentiated and polarized monolayer, which was co-cultured with HMC-1. In parallel, Caco2 cells were also co-cultured in IMDM medium without HMC-1 as blank (labeled as Caco2) to exclude the possibility that the medium change already causes a change in paracellular permeability. The integrity of Caco2 monolayers was affected by HMC-1 cells treated with CRH (500 nM) or JM25-1 (31.25 μM), and measured with an epithelial volt-ohm meter (Millicell^®^ ERS-2, Merck, Darmstadt, Germany). During the subsequent 3 h, the TEER value was monitored per to evaluate the resistivity. The resistivity was calculated by the formula: Resistivity (Ω·cm^2^) = (Ohm − Ohm_0_) × A; Ohm_0_: resistance value of the insert with culture medium only; Ohm: resistance value of the insert with cells; A: surface area of insert.

In order to assess epithelial permeability, FITC-dextran 4000 (FD4, 4 kDa, 100 μg/mL) was used to polarized monolayer after 3 h, and we detected the fluorescence intensity that penetrate into the lower layer by Nivo 3S plate reader (PerkinElmer, Waltham, MA, USA) with an excitation wavelength at 488 nm and an emission wavelength of 525 nm. 

### 4.4. N-Acetyl-β-D-Hexosaminidase Release Assay

To observe degranulation of MC, β-hexosaminidase release was evaluated as described earlier [48]. Briefly, HMC-1 and RBL-2H3 cells (1 × 10^5^ cells/well in 24-well plates) were incubated with Tyrode’s solution (8.0 g/L NaCl, 0.2 g/L KCl, 0.26 g/L MgSO_4_·7H_2_O, 0.065 g/L NaH_2_PO_4_·2H_2_O, 1.0 g/L Na_2_CO_3_, 0.2 g/L CaCl_2_, 1 g/L glucose) for 30 min, and the supernatant was discarded after centrifugation (1000 rpm, 5min). Compound 48/80 (10 μg/mL) was added and JM25-1 (7.8125–125 μM in Tyrode’s solution)-treated cells for another 10 min. Then, blank control cells were lysed with 0.1% Triton X-100 for 5 min to obtain the intracellular β-hexosaminidase. The supernatants (100 μL) were combined with 100 μL of 5 mM 4-nitrophenyl N-acetyl-β-D-glucosaminide dissolved with 0.05 M citric acid/sodium citrate buffer (pH 4.5) for 90 min at 37 °C. Then, hydrolysis reaction was terminated by the addition of 150 μL stop buffer (0.1 M Na_2_CO_3_ / NaHCO_3_, pH 10.7) and the absorbance was measured at 405 nm by Microplate Reader (PerkinElmer, Waltham, MA, USA). β-hexosaminidase release was calculated by the formula: β-hexosaminidase release (%) = (OD _treated_ − _media_/OD _blank control_ − _media_ × 100%).

### 4.5. Target Prediction

The SwissTargetPrediction database (swisstargetprediction.ch/ accessed on 20 December 2021) and the PharmMapper platform (lilab-ecust.cn/Pharmmapper/ accessed on 20 December 2021) were used to obtain the JM25-1 related targets. Gene targets linked to JM25-1 activity were thereby identified as “drug targets”. Irritable Bowel Syndrome (IBS) relevant targets were obtained as “disease targets” from the GeneCards (genecards.org/ accessed on 20 December 2021) and DisGeNET (disgenet.org/ accessed on 20 December 2021) databases.

### 4.6. Protein–Protein Interaction (PPI) Network Construction and Hub Target Screening

The STRING 11.0 database (string-db.org accessed on 20 December 2021) was used to appraise the interactions of the targets in a network with the species defined as “Homo sapiens”. Then, the PPI data was input into Cytoscape 3.7.1 software for visibility optimization and topology analysis.

### 4.7. Molecular Docking Simulation

To observe the direct binding of JM25-1 with the hub targets identified in the PPI network, we derived the 3D JM25-1 structure and crystal structures of the AKT1 protein from the PubChem and RCSB Protein Data Bank (rcsb.org/ accessed on 5 May 2022) databases, respectively. Surflex-Dock (SFXC) was used as the docking mode after protein structure preparation. Surflex-Dock scores (total scores) represent binding affinities.

### 4.8. Gene Ontology (GO) Term and Kyoto Encyclopedia of Genes and Genomes (KEGG) Pathway Enrichment Analyses

GO term and KEGG pathway enrichment analyses were performed to decipher the functions of differentially expressed genes, which were also used to predict the potential mechanism of drug action. We entered the gene symbols of the common targets into the Database for Annotation, Visualization and Integrated Discovery (DAVID) 6.8, a database with integrated GO and KEGG modules. 

### 4.9. Animals and Chronic Stress Experimental Design

Specific pathogen free (SPF) Wistar rats (8 weeks old, male, 180–220 g) were supplied by Chengdu Dashuo Experimental Animal Co. Ltd. (Chengdu, Sichuan, China). All animals were treated humanely in accordance with the requirements of the National Institutes of Health Laboratory Animal Care and Use Guidelines (8th Edition). Rats were housed with free access to commercial water and diet at an invariable room temperature (23 °C) under a 12 h light/dark cycles. After seven days of adaptive culture, rats were evenly divided into three groups according to body weight (Control group, Stress group, and Stress + JM25-1 group, *n* = 7). Chronic and acute exposure of rats to psychological stress was the model adopted in this experiment [7,49,50]. Briefly, as shown in Figure 4A, animals were placed on a platform (Diameter 8.9 cm, height 11.7 cm) positioned in the middle of a cage (65 × 47 × 41 cm) filled with water to 1 cm below the platform between 9:00 am and 10:00 am for 1 h daily for 10 consecutive. Control group was placed on the same platform above a waterless cage. Six hours later, the rats except control group were subjected to additional one pressure per day, including ultrasonic noise interference for 1 h, overnight in wet bedding, tilting the cage 45° for 12 h, 12 h without water, 12 h with starvation, 2 min with tail clips, or standing on ice for 10 min. Control rats did not need additional pressure. After seven consecutive days, rats under stress were intraperitoneal injected with JM25-1 (15 mg/kg, *n* = 7, dissolved in normal saline with 5% DMSO) and the other two groups were injected with normal saline with 5% DMSO. The drug was administered continuously for three days, and JM25-1 was administered 1 h before pressure every day. This experiment was approved by the ethical committee at The Ethics Committee of Southwest Jiaotong University Approval (Agreement No. SWJTU-2013-026).

### 4.10. Transplantation of Colonic Mucus from IBS Patient

The study protocol was approved by The Ethics Committee of Southwest Jiaotong University Approval (Agreement No. SWJTU-2013-026), and all patients were asked to sign a written informed consent and IBS-symptomatic assessment. The diagnosis of IBS is based on typical clinical symptoms that fulfilled the Rome IV criteria [51]. Briefly, IBS patients were included with abdominal pain relieved by defecation or associated with its onset, with a change in stool frequency (either an increase or decrease) or a change in the appearance of the stool (to either loose or hard). Patients with a family history of organic gastroenterological diseases and organic or severe psychiatric disorders that may lead to gastrointestinal symptoms were excluded. In total, six patients with IBS were recruited into this research. Under colonoscopy, colonic mucus of each individual was obtained with N-acetylcysteine washing at Z sigmoid.

Specifically, Wistar rats were divided into two groups (Mucus group and Mucus + JM25-1 group, *n* = 6), and anesthetized with phenobarbital after undergoing starvation for 24 h. Afterwards, IBS mucus was transplanted into rats’ colon of two groups using a 1 mm diameter polytetrafluoroethylene capillary. Transplanting position of each rat remained the same, which was 3 cm away from the anus. After 30 min, JM25-1 dissolved in normal saline with 5% DMSO was intraperitoneally injected into rats to Mucus + JM25-1 group, and the Mucus group was injected with solvent without JM25-1. Rats were sacrificed and colonic tissue was collected after overnight. 

### 4.11. Western Blot Analysis

After drug treatment, cell/small intestine tissue was harvested and lysed with 1 × RIPA lysis buffer (Beyotime, Shanghai, China) containing phenyl-methyl-sulfonyl fluoride (PMSF, Beyotime, Shanghai, China). The proteins were obtained from cell lysates and separated on SDS-PAGE. After transferring proteins to PVDF membranes (Milipore, Darmstadt, Germany), the membranes were blocked with TBST supplemented with 5% skimmed milk (Biofroxx, Guangzhou, China) for 1 h at room temperature. The membranes were then incubated with the primary antibodies (TNFα, IL-1β, CRHR1, CRHR2, GAPDH, p-PI3K, PI3K, p-AKT, AKT, p-mTOR, mTOR, and ZO-1; 1:1000 dilution, Proteintech, Wuhan, Chian) at 4 °C overnight, followed by an incubation with HRP-conjugated secondary antibody (Proteintech, Wuhan, China) for 1 h at room temperature. Signals were detected using the ECL Western blotting substrate (Millipore, Darmstadt, Germany) and quantified with imageJ software (ImageJ 1.46r, National Institutes of Health, Bethesda, MD, USA). The data were obtained from three independent experiments. 

### 4.12. Total RNA Isolation and Quantitation

Small intestine tissue or HMC-1 cells with a density of 5 × 10^5^/mL incubated in 6-well plates for 24 h were extracted total RNA using TRIzol reagent (Invitrogen, Carlsbad, CA, USA). Reverse transcription was performed using a ChamQ Universal SYBR qPCR Master Mix and quantitative PCR was performed with HiScript QRT SuperMix for qPCR (Vazyme, Nanjing, Jiangsu, China). Primer sequences used are listed as Table 2. The level of mRNA expression was calculated using the 2^−(∆∆C^_T_^)^ method and GAPDH was used as a loading control. 

### 4.13. Determination of Cytokine Secretion

Serum was acquired from rat tail vein centrifuged at 3000× *g* for 5 min. Supernatant level of diamine oxidase, D-lactic acid and bacterial endotoxin was detected according to the instructions. In brief, 20 microliters of sample were added into the testing panel for 20 min at 37 °C, and detected concentrations by Intestinal Barrier Biochemical Index Analysis System (Zhongshengjinyu, Beijing, China).

### 4.14. Immunofluorescence (IF)

There are two major types of immunofluorescent staining method: (1) Cells that attached to coverslips after experimental treatment were fixed with 4% paraformaldehyde for 20 min, and permeability with 0.2% Triton X-100 and blocked with PBS containing 5% BSA for 60 min in the next. The coverslips were followed by incubating with primary ZO-1/CRHR1/CRHR2, and labeled with specific secondary antibody for 1 h, and stained with DAPI (Biosharp, Hefei, China) for 5 min. The slides were washed with PBS and mounted by FluorSave^TM^ mounting media (Merck, Darmstadt, Germany). (2) The intestinal tissue was fixed in 10% phosphate-buffered formalin for immunofluorescent studies before paraffin embedding. For immunofluorescent staining of ZO-1 (1:500, Proteintech, Wuhan, Chian), CRHR1 (1:50, Proteintech, Wuhan, Chian), CRHR2 (1:200, Proteintech, Wuhan, Chian) and Tryptase (1:100, Santa Cruz, CA, USA), paraffin sections (5 μm) were dewaxed to rehydrate. After blocking with 10% goat serum, sections were incubated with primary antibody at 4 °C overnight, and detected with IgG (H + L) Highly Cross-Adsorbed Secondary Antibody (Thermofisher, Shanghai, China) for 1 h at room temperature, respectively. Representative images were captured under a fluorescence microscope and analyzed with imageJ Software (ImageJ 1.46r, National Institutes of Health, Bethesda, MD, USA).

### 4.15. Immunohistochemistry (IHC)

Colons were fixed in 10% formalin, embedded in paraffin, sectioned (5 μm), and deparaffinized. Subsequently, sections performed heat-induced antigen retrieval with Citrate-EDTA Antigen Retrieval Solution for 20 min (Beyotime, Shanghai, China) and stained with primary antibodies (TNFα, ZO-1). Thereafter, enhanced enzyme labeled secondary antibody (Zhongshan Goldenbridge, Beijing, China) and DAB Substrate was employed. Representative images were captured under fluorescence microscope (Leica Microsystems, Wetzlar, Germany).

### 4.16. Transmission Electron Microscopy (TEM)

The colon of transplanted rat was successively fixed in 3% glutaraldehyde and 1% osmium tetroxide, then dehydrated in acetone, permeated in epoxy 812 for a longer time, and embedded. Using a diamond knife to cut the semithin sections which stained with methylene blue, ultrathin sections were obtained. Afterwards, the sections were stained with uranyl acetate and lead citrate to examine with JEM-1400-FLASH Transmission Electron Microscope (JEOL, Tokyo, Japan).

### 4.17. Statistical Analysis

All data are analyzed by one-way univariate analysis of variance (ANOVA) using GraphPad Prism among the groups 8.0.1 (GraphPad Software, Inc., La Jolla, CA, USA). *^*^ p ≤* 0.05, *** p ≤* 0.01, **** p ≤* 0.001 are considered as highly statistically significant.

## Figures and Tables

**Figure 1 pharmaceuticals-16-00047-f001:**
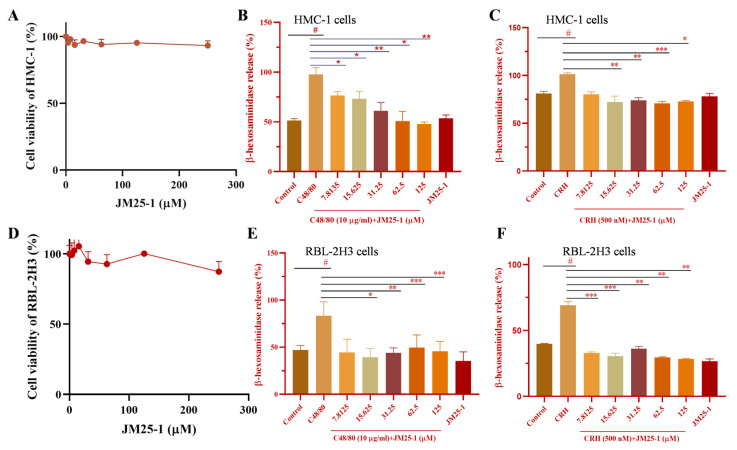
JM25-1 inhibits MC degranulation at safe concentration. (**A**) HMC-1 cells were treated with JM25-1 (0, 3.9, 7.8125, 15.625, 31.25, 62.5, 125, 250 μM) for 24 h and the cell viability was detected by CCK-8 assay. HMC-1 were pretreated with JM25-1 for 10 min (**B**) in C48/80 or (**C**) CRH-induced degranulation for 15 min. (**D**) RBL-2H3 cells were treated with JM25-1 (0, 3.9, 7.8125, 15.625, 31.25, 62.5, 125, 250 μM) for 24 h and the cell viability was detected by CCK-8 assay. RBL-2H3 were pretreated with JM25-1 for 10 min (**E**) in C48/80 or (**F**) CRH-induced degranulation for 15 min. The data are presented from at least three independent experiments run in triplicate. Bars, S.D. ^#^
*p* ≤ 0.05 vs. the Control; * *p* ≤ 0.05, ** *p* ≤ 0.01, *** *p* ≤ 0.001 vs. the C48/80 or CRH group.

**Figure 2 pharmaceuticals-16-00047-f002:**
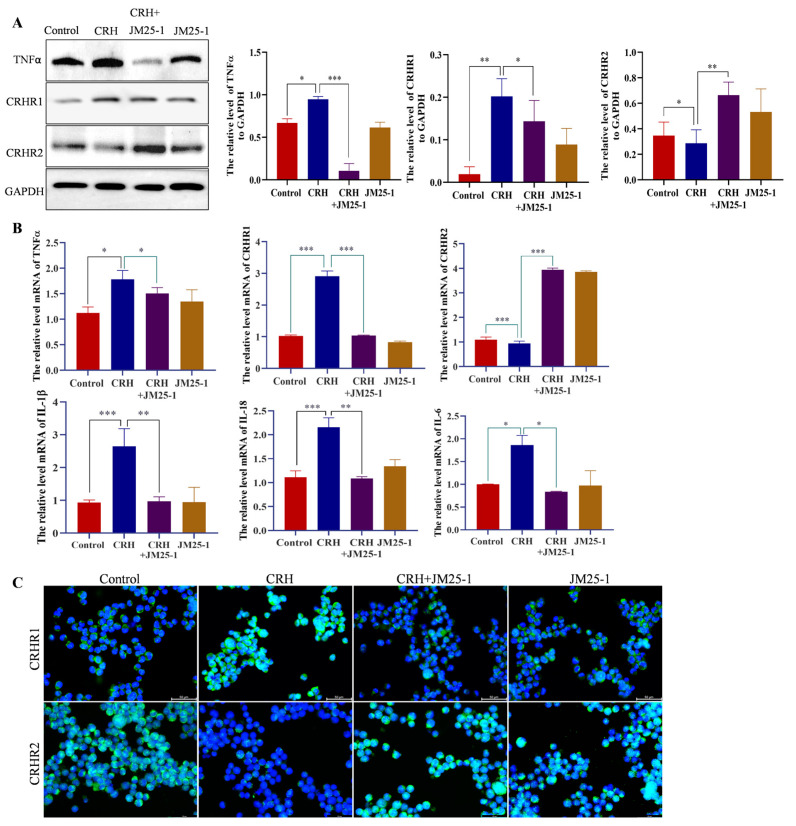
JM25-1 regulates the production of cytokine for CRHR1 and CRHR2. JM25-1 (31.25 μM) were treated in CRH (500 nM) -induced HMC-1 cells for 24 h, (**A**) the protein level of TNFα, CRHR1, and CRHR2 was detected by Western blot assay. The bar chart indicates the relative density of target protein to GAPDH; bars, S.D. * *p* ≤ 0.05, ** *p* ≤ 0.01, *** *p* ≤ 0.001. (**B**) After treatment, the mRNA level of TNFα, CRHR1, CRHR2, IL-1β, IL-18 and IL-6 were analyzed by real time PCR; bars, S.D. * *p* ≤ 0.05, ** *p* ≤ 0.01, *** *p* ≤ 0.001. (**C**) Immunofluorescence staining of CRHR1 and CRHR2 in HMC-1. Bar = 50 µm, magnification 400×. These data are presented from at least three independent experiments.

**Figure 3 pharmaceuticals-16-00047-f003:**
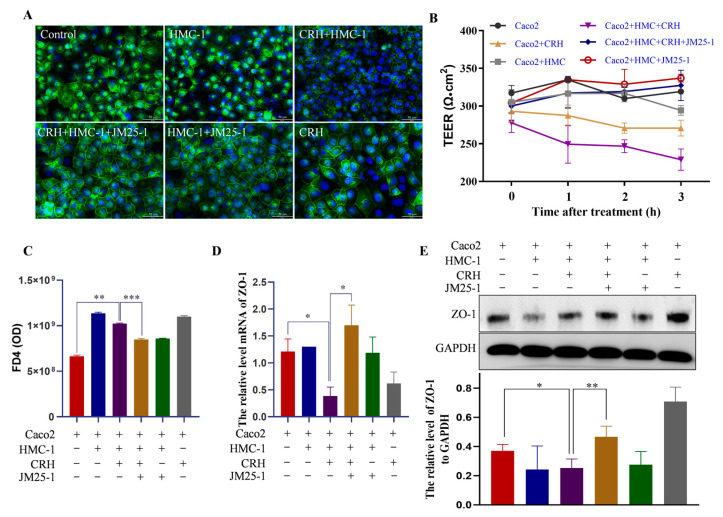
JM25-1 reduces the paracellular permeability of Caco2 cells via HMC-1 cells. HMC-1 cells were co-cultured with Caco2 that established a differentiated and polarized monolayer. JM25-1 were pretreated in HMC-1 for 30 min and then CRH stimulated for 24 h. The TJ were performed with (**A**) immunofluorescence staining of ZO-1; bar = 50 µm, magnification 400×; (**D**) mRNA of ZO-1; (**E**) protein level of ZO-1 in Caco2 cells monolayers. For detection of paracellular permeability, (**B**) transepithelial electrical resistance (TEER) at 0 h was measured before CRH treatment. After CRH treatment, TEER was monitored per hour until the trend is stable at 3 h; then, (**C**) the flux of FITC-dextran was examined; Bars, S.D. * *p* ≤ 0.05, ** *p* ≤ 0.01, *** *p* ≤ 0.001. *n* = 3 from three independent experiments.

**Figure 4 pharmaceuticals-16-00047-f004:**
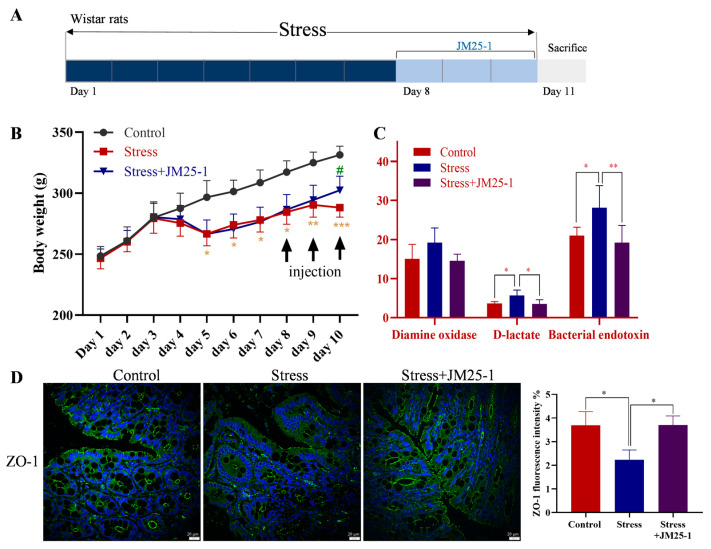
JM25-1 repairs intestinal permeability in stress induced rats. (**A**) Modeling method. (**B**) Body weight of rats after treatment with JM25-1 (i.p. every day, 15 mg/kg, *n* = 7); bar, S.D. * *p* ≤ 0.05, ** *p* ≤ 0.01, *** *p* ≤ 0.001 vs. the Control group. ^#^
*p* ≤ 0.05 vs. the Stress group (**C**) the level of diamine oxidase, D-lactic acid and bacterial endotoxin in serum on day 11. bars, S.D. * *p* ≤ 0.05, ** *p* ≤ 0.01; (**D**) immunofluorescence staining of ZO-1 in intestine. Bar = 20 µm, magnification 200×. bars, S.D. * *p* ≤ 0.05.

**Figure 5 pharmaceuticals-16-00047-f005:**
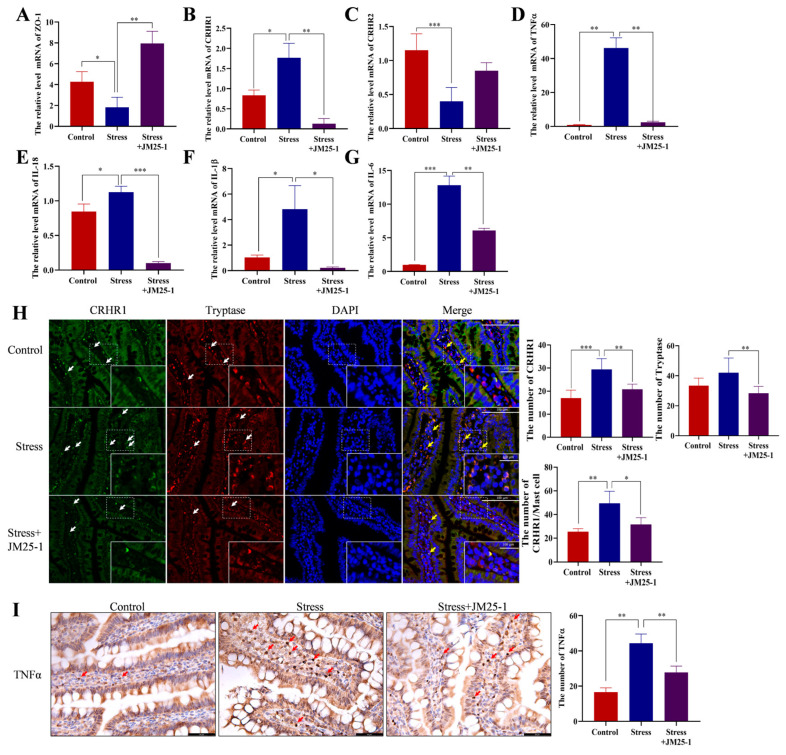
Effect of JM25-1 on the expression of CRH receptors and cytokines in small intestine of stress-induced rats. The relative expression of related mRNA in small intestine of stress-induced rats for (**A**) ZO-1, (**B**) CRHR1, (**C**) CRHR2, (**D**) TNFα, (**E**) IL-18, (**F**) IL-1β and (**G**) IL-6. The data are repeated in at least three independent experiments in triplicate compared with Control group. Bars, S.D. ** p ≤* 0.05, *** p ≤* 0.01, **** p ≤* 0.001. (**H**) The immunofluorescence staining of CRHR1 in Tryptase-labeled MC. The white arrows indicate representative CRHR1, Tryptase positive cells, respectively. Yellow arrows indicate the cells with FITC-CRHR1 located HRP-Tryptase. Magnification: 400× and 800×. Scale bar: 100 μm. Bar chart indicates the number of CRHR1, Tryptase labeled MC and CRHR1 to MC; bars, S.D. ** p ≤* 0.05, *** p ≤* 0.01, **** p ≤* 0.001. (**I**) The immunohistochemistry staining of TNFα in small intestine. Red arrows indicate distribution of TNFα. Magnification: 400×. Scale bar: 50 μm. Bars, S.D. *** p ≤* 0.01.

**Figure 6 pharmaceuticals-16-00047-f006:**
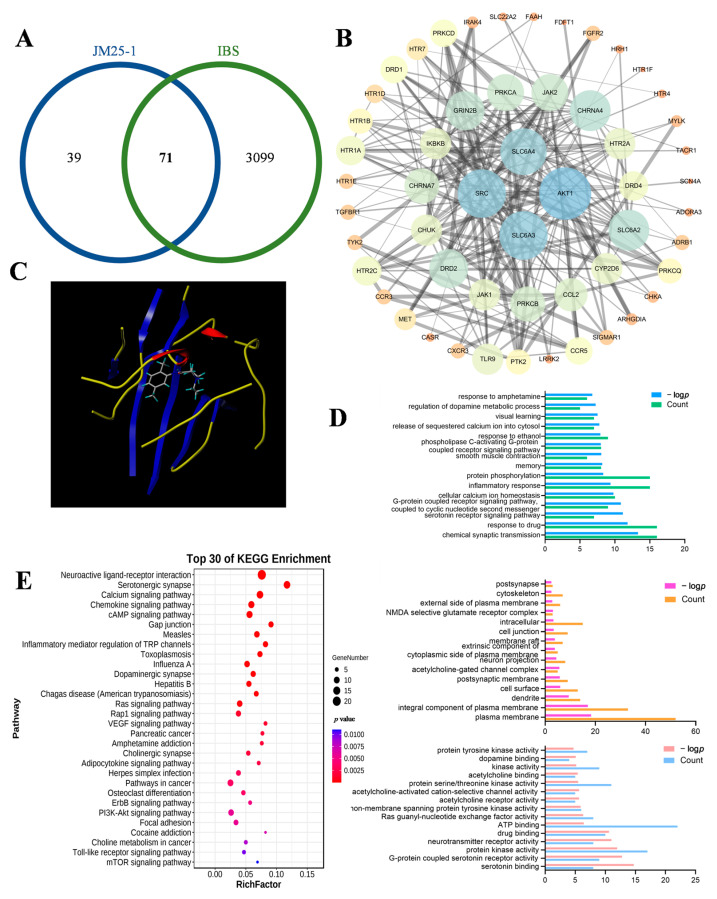
The effect of JM25-1 on IBS was discussed based on network pharmacology. (**A**) Venn diagram of JM25-1 targets and IBS targets. (**B**) Protein–protein interaction (PPI) network of bioactive compounds of JM25-1 against IBS. (**C**) Visualization of molecular docking between JM25-1 and IBS. (**D**) GO enrichment analysis of core target for JM25-1 against IBS, including biological process, cellular components and molecular function. (**E**) Top 30 enriched KEGG pathways of core target for JM25-1 against IBS. The color scales indicate the different thresholds of adjusted *p*-values, and the sizes of the dots represent the gene count of each term.

**Figure 7 pharmaceuticals-16-00047-f007:**
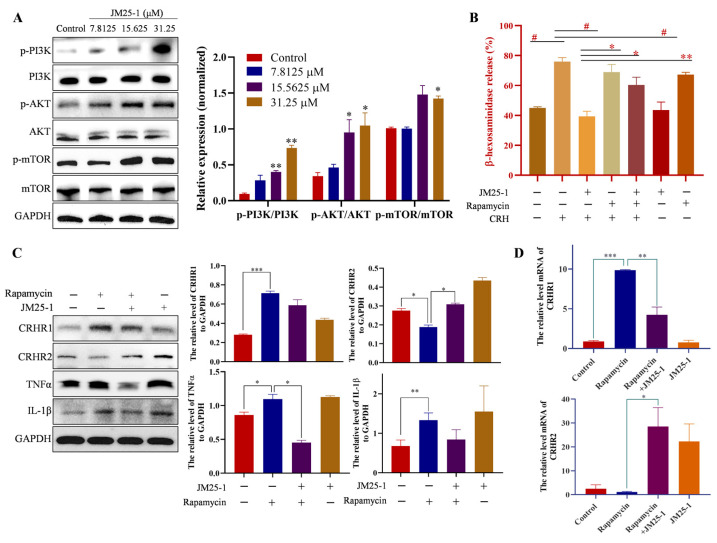
JM25-1 reduces degranulation and cytokine through the PI3K/AKT/mTOR/CRHR signaling pathway. HMC-1 cells were treated with JM25-1 at continuous concentrations for 24 h. (**A**) Western bolt was used to analyze the expression of p-PI3K, PI3K, p-AKT, AKT, p-mTOR, mTOR and GAPDH. Bars, S.D. * *p* ≤ 0.05, ** *p* ≤ 0.01, VS. the Control. (**B**) Cells treated with JM25-1 (31.25 μM) and Rapamycin (400 nM) regulate the release of β-hexosaminidase in CRH (500 nM) induced HMC-1. bars, S.D. ^#^
*p* ≤ 0.05, vs. the CRH group; * *p* ≤ 0.05, ** *p* ≤ 0.01, vs. the CRH + JM25-1 group. (**C**) HMC-1 was treated with JM25-1 or co-treated with Rapamycin for 24 h. The protein level of CRHR1, CRHR2, TNFα, IL-1β and GAPDH was analyzed by Western blot. (**D**) And the mRNA level of CRHR1 and CRHR2 was detected by real-time PCR. Bar chart indicates the relative expression of target proteins to GAPDH. bars, S.D. * *p* ≤ 0.05, ** *p* ≤ 0.01, *** *p* ≤ 0.001, vs. the Rapamycin group.

**Figure 8 pharmaceuticals-16-00047-f008:**
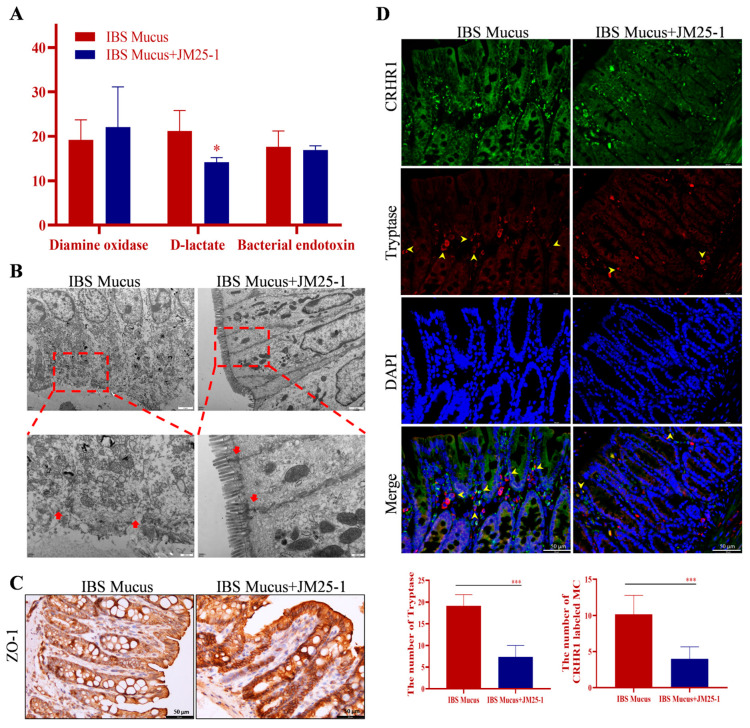
JM25-1 regulated intestinal barrier defect in IBS intestinal mucus transplantation rats. (**A**) The level of diamine oxidase, D-lactic acid and bacterial endotoxin in serum. bars, S.D. * *p* ≤ 0.05. (**B**) Transmission electron microscope images of colonic segments. Magnification: 8000×. Scale bar: 2 μm. Magnification: 25,000×. Scale bar: 500 nm. (**C**) The immunohistochemistry staining of ZO-1 in the colon. (**D**) The immunofluorescence staining of CRHR1-labeled Tryptase. Magnification: 400×. Scale bar: 50 μm. Bar chart indicates the number of Tryptase and CRHR1 labeled MC; bars, S.D. *** *p* ≤ 0.001.

**Table 1 pharmaceuticals-16-00047-t001:** Details of core targets of JM25-1 against IBS.

Name	Betweenness Centrality	Closeness Centrality	Degree
AKT1	0.2215193	0.58888889	21
SLC6A3	0.14536581	0.55789474	19
SRC	0.13163614	0.54639175	19
SLC6A4	0.10762354	0.47321429	18
SLC6A2	0.04392008	0.43442623	15
CHRNA4	0.04196715	0.5047619	15
DRD2	0.08383962	0.53535354	14
GRIN2B	0.05834103	0.54639175	14

**Table 2 pharmaceuticals-16-00047-t002:** List of primer.

Name	Primer: 5′ to 3′
TNFα-Forward (Human)	CCAGGGACCTCTCTCTAATCA
TNFα-Reverse (Human)	TCAGCTTGAGGGTTTGCTAC
CRHR1-Forward (Human)	TGGTGTCCGCTACAATACCA
CRHR1-Reverse (Human)	AGTGGCCAGGTAGTTGATG
CRHR2-Forward (Human)	CCGGAATGCCTATCGAGAATG
CRHR2-Reverse (Human)	GGTCATACTTCCTCTGCTTGTC
IL-1β-Forward (Human)	ATGACCTGAGCACCTTCTTTC
IL-1β-Reverse (Human)	TGCACATAAGCCTCGTTATCC
IL-18-Forward (Human)	CAGATCGCTTCCTCTCGCAA
IL-18-Reverse (Human)	CCAGGTTTTCATCATCTTCAGCTAT
IL-6-Forward (Human)	CCAGGAGAAGATTCCAAAGATGTA
IL-6- Reverse (Human)	CGTCGAGGATGTACCGAATTT
GAPDH-Forward (Human)	GTCAACGGATTTGGTCGTATTG
GAPDH-Reverse (Human)	TGTAGTTGAGGTC AATGAAGGG
ZO-1-Forward (Rat)	CTTGCCACACTGTGACCCTA
ZO-1-Reverse (Rat)	ACAGTTGGCTCCAACAAGGT
CRHR1-Forward (Rat)	GGTGGCCTTTGTCCTCTTCTT
CRHR1-Reverse (Rat)	AAAGCCGAGATGAGGTTCCA
CRHR2-Forward (Rat)	CATCATCCTCGTGCTCCTCAT
CRHR2-Reverse (Rat)	TGGAGGCTCGCAGTTTTGT
TNFα-Forward (Rat)	CGTAGCCCACGTCGTAGCA
TNFα-Reverse (Rat)	GTCTTTGAGATCCATGCCATTG
IL-1β-Forward (Rat)	TCAGGAAGGCAGTGTCACTCAT
IL-1β-Reverse (Rat)	AAGAAGGTGCTTGGGTCCTCAT
IL-6-Forward (Rat)	CCAGGAGAAGATTCCAAAGATGTA
IL-6-Reverse (Rat)	CGTCGAGGATGTACCGAATTT
IL-18-Forward (Rat)	GCCTCAAACCTTCCAAATCA
IL-18-Reverse (Rat)	TGGATCCATTTCCTCAAAGG
GAPDH-Forward (Rat)	GGGTGTGAACCACGAGAAATATG
GAPDH-Reverse (Rat)	CCACGATGCCAAAGTTGTCA

## Data Availability

All datasets generated for this study are included in Appendix A.

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
