# Peer review of "A Novel Mast Cell Stabilizer JM25-1 Rehabilitates Impaired Gut Barrier by Targeting the Corticotropin-Releasing Hormone Receptors"

_pharmaceuticals, 2022, doi:10.3390/ph16010047_

Round 1

Reviewer 1 Report

The manuscript of Sun et al.  “A Novel Mast Cell Stabilizer JM25-1 Rehabilitates Impaired 2 Gut Barrier by Targeting the Corticotropin-releasing Hormone 3 Receptors” addresses the possible therapeutic use of the nonfluorinated lidocaine analog JM25-1 in a disturbed intestinal barrier. Based on CRH-activated mast cells (MC) as a relevant cell type in stress-induced barrier impairment, they analyzed the impact of JM25-1 in vitro with MC and epithelial cells and in vivo with a chronic stress model. From their data the authors conclude that JM25-1 has a therapeutic capacity for intestinal barrier defects by targeting MC via PI3K/AKT/mTor signaling.  

Despite interesting questions, many points remain open in the work of Sun et al. Apart from the fact that the manuscript has some deficiencies (Figure 1 is missing, supplement folder has no content, material and methods section and Figure legends needs to be improved, controls are missing, lack of information on the number of animals used and how often each experiment was performed), the involvement of mast cells in the improvement of intestinal barrier defects induced by JM25-1 administration in vivo is not convincing. Furthermore, several questions arising from the data shown are not addressed.

Major points:

Figure 1 is missing

Figure 2:

-       Degranulation of mast cells is induced within minutes. Thus, analysis 24h after treatment seems to be pretty late. Did the authors also investigated other, in particular earlier time points?

-       It seems as DMSO control is missing, the method should be described more precisely

-       Fig.2A: application of JM25-1 seem to impact the cells already. How do the authors explain the increase of CRHR1 in the JM25-1 control on protein level?

-       B: did the authors also analyzed the protein level of IL-1b and IL-18? What about IL-6, a cytokine known to be released by activated mast cells and to be involved in chronic intestinal inflammation (Powell et al., Gastroenterology 2015; Parisinos et al., Gastroenterology 2018)?

-       C: the resolution of the images should be improved

Figure 3:

The description is somewhat irritating, according to the material and method section a transwell assay was performed, which is not clearly mentioned in the figure legend or the text. Since Caco2 and HCM-1 are cultured in different media, a control with Caco2 in HCM-1 medium (without HCM-1 cells) should be performed to exclude that the medium change already causes a change in paracellular permeability.

-       The control Caco2 +JM25-1 is missing

-       Fig.3B: what exactly does treatment mean in this figure – start of the coculture?

-       it would be helpful to use the same color code in the different parts of the figure

-       How do the authors explain the data for the condition Caco2 + CRH? The protein data in IF and western blot of ZO-1 differ, the RNA values do not behave in the same way as the protein data showing the highest ZO-1 protein level but at the same time reduced RNA level, decreasing TEER values and increased FD4. It seems as CRH already impacts Caco2 cells in the absence of MC, thus the author should also investigate the combination of CRH and JM25-1 on Caco2 without MC

-       It seems as if the magnification in the condition CRH differs from the rest, as the nuclei appear enlarged

Figure 4-6

These 3 figures could be combined in one, since they all seem to refer to the same experiment. Next to MC expression of CRHRs has been descripted for other cell types involved in the gut barrier disorders (reviewed e.g. in Chatoo et al., Frontiers in Immunology 2018). The data shown here do not allow the conclusion that JM25-1 directly targets MC.

Figure 4

-       The description of the experimental set up must be improved,

o   what exactly is meant by “model”? is it the same as “WAS”?

o   Is this an accepted model for chronic stress? which type of stress was exactly applied to the rats, was it the same for all rats in the group “model” and “JM25-1”?

o   in material and methods, the authors state that the JM25-1 injection started at day 7, in Figure 4A it is indicated at day 8

-       What was the intention to apply JM25-1 at day 8 and not earlier? The difference between “model” and “JM25-1” is only marginal, possibly the differences would be more pronounced with a longer application

-       How do the authors explain the increase in weight after day 5, have the animals adapted to the stress??

-       At which day were the analysis shown in Fig 4B performed? Did the authors also analyzed other time points?

Figure 5

-       Did the authors also include the analysis of MC-specific genes? This could provide hints that indeed MC are targeted by JM25-1

-        

Figure 6

-       The IF images are not convincing, the resolution should be increased and at least sections with stronger magnification should be shown

Figure 7 and 8 could be combined

Figure 9

-       Fig. 9B: The controls JM25-1 alone and rapamycin alone are missing

-       Fig. 9B: The application of DSCG is not explained nor commented in the results, only in the discussion.

-       Fig. 9C/D: In the figure legend and text the analysis of tryptase by Western blot is mentioned but not shown in the figure. How do the authors explain the increase of TNF-a and IL-1b by JM25-1 compared to control? The same question arises for CRHR1 (comparable to Fig. 2A), especially since this does not apply at the RNA level (9D). Similar, the upregulation of CRHR2 RNA by the combination of rapamycin and JM25-1 should be explained. If I understood it correctly, the authors state that rapamycin prevented the JM25-1 induced up-regulation of CRHR1 and JM25-1 induced down-regulation of CRHR2 (“after treating with Rapamycin in MC, and found that mTOR inhibitor prevented JM25-1 from the down-regulation of CRHR1 and up-regulation of CRHR2 (Figure. 9C-D)“, page 11. However, the protein data show an upregulation of both, CRHR1 an2 by JM25-1 compared to control, which is not reflected on RNA level.

Figure 10

-       Fig. 10D: the quality of the IF is poor, there is a strong background staining and no clear positive cells detectable

Reviewer 2 Report

This present study demonstrated the protecting role of JM25-1 against irritable bowel syndrome by targeting corticotropin-releasing hormone receptor. There are some major concerns in this study:

1) Very first concern is that entire figure 1 is missing in the manuscript. This shows the utter negligence.

2) In Fig 3E, why ZO-1 expression is reduced in Caco2 cells with HMC-1 without pretreatment with CRH?

3) In Fig 4A, there is no difference in model and JM25-1.

4) In Fig 4C, there is no visible difference in Model and JM25-1.

5) Similarly in Fig 6A, there is no significant difference among panels. Need to be replaced by better quality high magnification image.

6) In Fig 9C, why the effect of JM25-1 is similar to rapamycin?

Minor comment:

1) All the bars throughout the figure should be of same color code. Some are black and white and some color.

2) Line 87 (Result section 2.2) is repeated in line 127 179.

Manuscript need extensive revision and all the concerns should be addressed.

Round 2

Reviewer 1 Report

The revised version of Sun et al. has become more comprehensible and shows significant improvements, however, the main criticism remains: it has not been shown that mast cells are involved in the improvement of intestinal barrier defects induced by JM25-1 administration.

Although the authors can show an in vitro effect of JM25-1 on degranulation and expression of CRHR on MC, questions remain:

1.      Fig. 1: Usually, the amount of β-hexosaminidase released is calculated as a percentage of the maximum possible release in total lysis, which is - based on the cited publication and other referenced papers described by the formula: release = (supernatant - blank) / (total - blank) X 100% (Dastych et al.; J immunol 2007). Accordingly, a maximum of 100% of β-hexosaminidase can be released, but values above 100% are shown for C48/80 in Fig. 1B. – how is this possible?

2.      The data in Fig. 3 clearly show that addition of CRH already changes the levels of TEER, FD4, and ZO-1 expression in Caco2 cells. It has been shown that JM25-1 modulates different cell types such as smooth muscle cells, T-cells or eosinophils (Sera et al. 2016). As stated by the authors, it is already described that MC play a central role in intestinal permeability. Thus, to prove that the changes observed in Caco2 cells after addition of JM25-1 are due specifically to stabilization of MC, it must be excluded that JM25-1 does not act on Caco-2 cells in a MC-independent manner. This could be easily answered by the already requested analysis of the condition Caco2 cells in combination with CRH and JM25-1.

The authors have shown a protective effect of JM25-1 in their rat stress model. but as evidence for the involvement of mast cells is not convincing.  The authors refer to the immunofluorescence staining shown in (Fig. 5H), however, both stains show a strong background staining and neither for CRHR1 nor for Tryptase positive cells are clearly detectable. Consequently, it is not comprehensible how it comes to the selection of cells marked as MC in the overlay (Fig. 5H, right column). All other data shown in the in vivo model are not MC-specific.

Reviewer 2 Report

Authors have adequately responded to all the concerns. I would recommend the manuscript should be accepted for publication.

Round 3

Reviewer 1 Report

In their second revised version, Sun et al. still could not refute the main criticism that it was not convincingly shown that the protective effect of JM25-1 in vivo is through targeting of mast cells. The only direct evidence that JM-25 also acts on mast cells in vivo to attenuate intestinal barrier defects is based on immunofluorescence (Fig 5H, Fig 8D), in which it is not possible to understand at the magnification and resolution shown which cells were scored as positive and which were not. Since CRHR1 is not expressed only by mast cells and JM25-1 does not act only on mast cells, this is essential to demonstrate. The authors' arguments that the resolution meets the requirements of the journal so they do not have to improve it or that the staining was not successful with a lower background are not acceptable. For example, the authors could perform immunohistochemical staining and either detect CRHR1 and MC simultaneously with a double staining, or show CRHR1 once and MC once on two consecutive sections. They could identify mast cells by staining of Mcpt1 (e.g. Groll et al., Cancers 2022), by detecting MC chymase by naphthol AS-D chloroacetate esterase (NASDCE) activity (e.g. Cespedes et al., Front Immunol 2022), by immunohistochemistry of tryptase (e.g. Pons et al., Front Immunol 2017) or toluidine blue staining (e.g. Ribatti, Int Arch Allergy Immunol 2018). Additionally, they could analyze the expression of MC-specific genes. Expression of CRHR1 could be also assessed by immunohistochemistry (e.g. Liu et al., Endocrine-related cancer, 2014). Since other groups have published CRHR1 IF-staining without a strong background at least in mice, the protocol may also need to be adapted and/or a different magnification chosen (e.g. Garcia et al., Brain Struct Funct 2016). Furthermore, CRHR1 could be shown by in situ hybridization (e.g. Bender et al., PlosOne 2015).

Round 4

Reviewer 1 Report

With the revision of the immunofluorescence (Fig. 5H and 8D) it is now actually possible to recognize cells and thus better understandable how the evaluation was performed. As also noted by the authors, there is still high background staining, yet the figures are more convincing, especially with the section magnification.